# Comparison of Major Protein-Source Foods and Other Food Groups in Meat-Eaters and Non-Meat-Eaters in the EPIC-Oxford Cohort

**DOI:** 10.3390/nu11040824

**Published:** 2019-04-11

**Authors:** Keren Papier, Tammy YN Tong, Paul N Appleby, Kathryn E Bradbury, Georgina K Fensom, Anika Knuppel, Aurora Perez-Cornago, Julie A Schmidt, Ruth C Travis, Timothy J Key

**Affiliations:** 1Cancer Epidemiology Unit, Nuffield Department of Population Health, University of Oxford, Richard Doll Building, Old Road Campus, Oxford OX3 7LF, UK; Tammy.Tong@ndph.ox.ac.uk (T.Y.N.T.); paul.appleby@ndph.ox.ac.uk (P.N.A.); Georgina.Fensom@ndph.ox.ac.uk (G.K.F.); anika.knuppel@ndph.ox.ac.uk (A.K.); Aurora.Perez-Cornago@ndph.ox.ac.uk (A.P.-C.); julie.schmidt@ndph.ox.ac.uk (J.A.S.); ruth.travis@ndph.ox.ac.uk (R.C.T.); tim.key@ndph.ox.ac.uk (T.J.K.); 2National Institute for Health Innovation, The University of Auckland, Auckland 1072, New Zealand; k.bradbury@auckland.ac.nz

**Keywords:** vegetarians, vegans, low-meat, cohort, food intake, diet

## Abstract

Differences in health outcomes between meat-eaters and non-meat-eaters might relate to differences in dietary intakes between these diet groups. We assessed intakes of major protein-source foods and other food groups in six groups of meat-eaters and non-meat-eaters participating in the European Prospective Investigation into Cancer and Nutrition (EPIC)-Oxford study. The data were from 30,239 participants who answered questions regarding their consumption of meat, fish, dairy or eggs and completed a food frequency questionnaire (FFQ) in 2010. Participants were categorized as regular meat-eaters, low meat-eaters, poultry-eaters, fish-eaters, vegetarians and vegans. FFQ foods were categorized into 45 food groups and analysis of variance was used to test for differences between age-adjusted mean intakes of each food group by diet group. Regular meat-eaters, vegetarians and vegans, respectively, consumed about a third, quarter and a fifth of their total energy intake from high protein-source foods. Compared with regular meat-eaters, low and non-meat-eaters consumed higher amounts of high-protein meat alternatives (soy, legumes, pulses, nuts, seeds) and other plant-based foods (whole grains, vegetables, fruits) and lower amounts of refined grains, fried foods, alcohol and sugar-sweetened beverages. These findings provide insight into potential nutritional explanations for differences in health outcomes between diet groups.

## 1. Introduction

Vegetarian diets, characterized by the avoidance of meat, meat products and fish and vegan diets, characterized by abstention from all animal products, have become increasingly popular in Western countries [1]. There is also a growing body of epidemiological evidence regarding the health effects of vegetarian diets. Prospective cohort studies have reported that compared with meat-eaters, non-meat-eaters might have lower risks of obesity [2], ischemic heart disease [3], diverticular disease [4], cataracts [5] and some cancers [6], but higher risks of some fractures [7,8]. Studies that have investigated the health effects of substituting red meat with vegetarian protein sources have observed lower risks of non-alcoholic fatty liver disease [9], coronary heart disease [10], stroke [11] and total mortality [12].

Comprehensive information on the food consumption patterns of non-meat-eaters is needed to better understand the differences in health outcomes between diet groups. Some previous studies have described food intakes of vegetarians and vegans [13,14,15,16,17,18,19,20,21,22] and reported that, compared to meat-eaters, non-meat-eaters consumed more soy and other legumes, nuts and seeds, whole grains, vegetables and fruits and less sugary drinks, refined grains, fried foods and alcohol. However, some studies were based on small numbers of vegetarians or vegans [14,16,17,18,19,21] and only one study investigated vegetarian diets in the UK [15]. Therefore, there is a need for more information on this in studies with a large number of vegetarians to further characterize the food intakes of non-meat-eaters.

The European Prospective Investigation into Cancer and Nutrition (EPIC)-Oxford study collected detailed data on food intakes in a large cohort of meat-eaters and non-meat-eaters (45%) living throughout the UK. The aims of the current study are to describe and compare intakes of major protein-source foods and other food groups in regular meat-eaters, low meat-eaters, poultry-eaters, fish-eaters, vegetarians and vegans participating in the EPIC-Oxford study. 

## 2. Materials and Methods 

### 2.1. Study Participants

EPIC-Oxford is a prospective study of ~65,000 men and women aged 20+ at recruitment from across the UK. Participants were recruited between 1993 and 1999 through general practitioners (GPs) and postal questionnaires [2]. The questionnaire collected information on a wide range of topics including socio-demographic, diet, lifestyle and health factors. The analyses in the present study are based on dietary data collected in the third follow-up questionnaire, mailed to participants ~14 years (i.e., around 2010) after recruitment. The EPIC-Oxford study protocol was approved by a multicentre research ethics committee (Scotland A Research Ethics Committee) and all participants provided written informed consent.

### 2.2. Assessment of Diet and Diet Group

Participants completed a 112-item semi-quantitative food frequency questionnaire (FFQ), reporting foods consumed over the past 12 months. The FFQ was based on the validated baseline FFQ [23] and included additional vegetarian food items; it can be seen online [24]. Participants indicated the frequency of consumption of each food with responses ranging from ‘never’ to ‘6 or more times daily’. Mean daily intakes in grams were calculated by multiplying the frequency of consumption of each food by a standard portion size [25]. For major protein sources, mean daily energy intakes in kilocalories (kcal) were also calculated by multiplying the mean daily gram intakes by the energy content for the food item [26]. The 112 food items were divided into 45 food groups according to nutritional content (major source of protein or other food group) and definitions used in previous studies (Appendix A) [15]. As part of the FFQ, participants were asked to report whether they consumed meat, fish, dairy products or eggs, indicating how often they ate each of several types of meat, fish and dairy products where appropriate. Responses to these questions were used to calculate grams of meat intake and categorize participants into one of six diet groups: regular meat-eaters (those who consumed >50 grams (g) of total meat (any) daily); low meat-eaters (those who consumed <50 g of total meat (any) daily); poultry-eaters (those who consumed poultry but did not consume red meat); fish-eaters (those who consumed fish but did not consume meat); vegetarians (those who did not consume meat or fish but did consume dairy products, or eggs); and vegans (those who did not consume meat, fish, dairy products, or eggs).

### 2.3. Eligibility

Eligible participants were those who completed the third follow-up questionnaire and were aged under 90 years at third follow-up (*n* = 32,424). Of these, participants with an unknown diet group were excluded (*n* = 47), as were those with incomplete or implausible dietary data (*n* = 2138), (defined as having an estimated daily energy intake of less than 3349 kJ (800 kcal) or more than 16,747 kJ (4000 kcal) for men and less than 2093 kJ (500 kcal) or more than 14,654 kJ (3500 kcal) for women), missing energy intake data, or more than 20% of relevant food items missing in the FFQ. Accordingly, the analysed cohort included 30,239 participants, including 23,875 women and 6364 men.

### 2.4. Statistical Analysis

Sociodemographic, lifestyle and health characteristics taken from the baseline questionnaire (and the third follow-up where available) were compared across the six diet groups. We then calculated age-adjusted mean intake in grams (g) of all 45 food groups for each of the diet groups. We also standardised the age-adjusted mean intakes in grams to a 2000 kcal/day diet (calculated by dividing each participant’s food group intake by their daily energy intake in kcal and multiplying it by 2000 kcal); and calculated the proportion of energy contributed by each major protein food (calculated by dividing the energy intake in each major protein food group by total energy). Analysis of variance (ANOVA) was used to test for overall differences between the age-adjusted mean intakes across all six diet groups. Pairwise comparisons with Bonferroni correction were used to determine the statistical significance of differences between the regular meat-eaters and all other groups. Statistically significant differences between diet groups are reported in the text. We also calculated the relative and age-adjusted mean consumption of food groups for low and non-meat-eaters compared to regular meat-eaters, as the ratio of the age-adjusted mean intake in low and non-meat-eaters and the age-adjusted mean intake in the regular meat-eaters. All analyses were carried out using Stata (version 15.0). All statistical tests were two-sided with P values <0.05 considered statistically significant.

## 3. Results

Table 1. shows the participant characteristics by diet groups for men and women. Approximately one-third of men and one-quarter of women were vegetarian or vegan. Overall, compared with regular meat-eaters, low meat-eaters, poultry-eaters and non-meat-eaters were younger, had a higher education level, a lower socio-economic status, were less likely to smoke and consume alcohol, had higher levels of physical activity, a lower BMI, and, expressed as percentage of total energy intake, higher intakes of carbohydrates and lower intakes of protein and fat.

Mean intakes of major protein-source foods among meat-eaters and non-meat-eaters are shown in Table 2 (men) and Table 3 (women). The p values for differences between diet groups for all major protein-source foods were less than 0.0001, indicating that the foods consumed by the diet groups differed significantly. By definition, vegetarians and vegans did not consume red meat, processed meat, poultry or fish and, as expected, consumed more plant-based protein sources including legumes and other vegetarian protein alternatives (i.e., tofu, soya, Quorn) compared with meat-eaters. Vegetarians consumed the most cheese and vegans consumed the highest quantities of plant milk and nuts; more than 2.5 times the amount consumed by the regular meat-eaters. This pattern was similar across both the non-standardised and the 2000 kcal/day standardised intakes, showing that the differences between the diet groups for major protein sources were largely unrelated to differences in total energy intake.

Regular meat-eaters consumed nearly a third of their total energy intake from high protein-source foods (meat and fish: 15%; dairy and plant milk: 6%; and cheese, yogurt and eggs: 6%). Vegetarians consumed about a quarter of their total energy intake from high protein-source foods (legumes, nuts and vegetarian alternatives: 11%; cheese, yogurt and eggs: 8%; and dairy and plant milk: 5%). Vegans consumed approximately a fifth of their total energy intake from high protein-source foods (legumes, nuts and vegetarian alternatives: 18%; and plant milk: 5%) (Appendix A).

Intakes of other food sources by diet group are shown in Table 4 (men) and Table 5 (women). Overall, low and non-meat-eaters consumed higher amounts of vegetables and whole grain foods and lower amounts of fried foods, refined grains and sugary drinks than regular meat-eaters. Low meat-eaters, poultry-eaters, fish-eaters, vegetarians and vegans each consumed significantly more brown rice and couscous (with vegans consuming approximately double), and significantly less fried or roasted potatoes and coffee than regular meat-eaters. Low meat-eaters, fish-eaters, vegetarians and vegans consumed significantly less boiled potatoes, white bread and ice cream and significantly more wholemeal bread and wholemeal pasta than regular meat-eaters (with vegans consuming approximately double). Low meat-eaters, fish-eaters and vegetarians consumed significantly less fruit squash and spirits compared with regular meat-eaters. Low meat-eaters, vegetarians and vegans consumed significantly less white rice than regular meat-eaters. Low meat-eaters consumed significantly less milk desserts than regular meat-eaters. Fish-eaters, vegetarians and vegans consumed significantly more vegetables and soy desserts than regular meat-eaters. Fish-eaters and vegetarians consumed significantly more other bread and pizza and significantly less soft drinks and diet drinks compared with regular meat-eaters. Fish-eaters consumed significantly more crisps than regular meat-eaters. Vegetarians and vegans consumed significantly less wine compared with regular meat-eaters. Vegans consumed significantly less white pasta, pizza and tea and significantly more fruit compared with regular meat-eaters. Adjusting for education and socio-economic status had minimal influence on these results. Additional sex-specific differences between the diet groups are described in Appendix A.

The relative mean consumption of food groups for low and non-meat-eaters compared to regular meat-eaters, after adjustment for age, is shown in Figure 1 for men and Figure 2 for women. Compared with regular meat-eaters, fish-eaters, vegetarians and vegans consumed more than double the amounts of legumes, vegetarian alternatives and nuts. Among men, vegetarians and vegans consumed 1.5 times as much the sum of brown rice, wholemeal pasta, brown and wholemeal bread than regular meat-eaters, whereas only vegans consumed this amount among women.

## 4. Discussion

This study assessed intakes of major protein-source foods and other foods in different groups of meat-eaters and non-meat-eaters living in the UK. Our results indicate that there are large differences in dietary intakes between meat-eaters and non-meat-eaters; non-meat-eaters consumed higher amounts of soy, legumes, pulses, nuts and seeds, whole grains, vegetables and fruits, and lower amounts of refined grains, fried foods, alcohol and sugar-sweetened beverages (SSBs). These results were similar when we standardised intakes to a 2000 kcal daily diet, indicating that our findings were largely independent of energy intakes. 

Our finding of a higher consumption of a wide variety of plant-based foods in low and non-meat-eaters is consistent with findings from previous studies. The Adventist Health Study-2 and the UK Biobank both observed higher intakes of legumes, vegetarian protein alternatives (e.g., soy, tofu), nuts, whole grains, vegetables and fruits among low and non-meat-eaters [15,20]. Likewise, the NutriNet-Santé study in France and the Netherlands cohort study both reported a higher consumption of soy, cereals or grains, legumes or pulses, nuts, vegetables and fruits among non-meat-eaters [13,18]. Similar findings were reported by smaller studies [14,16,17,19,21,22].

It might be expected that vegetarians and vegans would replace meat with higher intakes of animal-sourced protein alternatives (including dairy and eggs) and non-animal protein alternatives (including legumes and nuts), respectively. However, our findings suggest that vegetarians and vegans did not completely replace meat consumption with non-meat protein sources and high-protein plant-sources but increased their consumption of a large variety of plant-based foods and consumed lower amounts of high protein-sourced foods compared with meat-eaters (proportion of total energy from high protein-sourced foods was one-third in regular meat-eaters, one-quarter in vegetarians; and one-fifth in vegans). Relatively low protein intakes have been previously observed in vegetarians and vegans in this cohort [27]. For vegans, we noted a higher consumption of plant milk and nuts, but also the highest consumption of brown rice, wholemeal pasta, couscous and wholemeal bread. This has also been observed in previous studies [15,20]. For vegetarians, we found lower intakes of total dairy and egg consumption compared with meat-eaters. However, cheese consumption was the highest in vegetarians. This pattern of dairy consumption has been reported previously [15,18,19,22]. Cheese can be high in energy, so it is possible that, to achieve energy requirements in their diet, vegetarians might preferentially replace meat with cheese over other lower calorie dairy products. The findings for egg consumption are less consistent in the literature [13,15,18,20,21,22] and we found that egg consumption was low in all diet groups. It is possible that in this ‘health conscious’ cohort [27], low egg consumption is due to the perceived healthfulness of plant-based foods, and thus high-protein vegetarian alternatives (including legumes, soy and nuts) and other plant-based foods (e.g., whole grains) are the preferred food substitutes for meat among vegetarians and vegans. 

We observed a lower consumption of refined carbohydrates, fried foods, alcohol and foods high in free sugars (e.g., ice cream and SSBs) among low and non-meat-eaters. Similar findings were reported by the largest previous studies that also found lower consumption of fried potatoes [15,20], refined grains, sweet and fatty foods, sugary drinks and alcoholic beverages [13,15,20] among non-meat-eaters. These findings suggest that non-meat-eaters might be consuming an overall “healthier” diet than meat-eaters. 

Well-planned vegetarian and vegan diets can comply with national dietary recommendations [28] and our study, together with estimates of nutrient intakes in previous studies [27,29], supports this. Compared to meat-eaters, the non-meat-eaters in this study consumed a diet that was consistent with most of the UK’s food recommendations, i.e., high in a wide variety of plant-based foods, vegetables and fruits and low in red and processed meat, refined grains, sugary foods and alcohol [30]. 

Important strengths of this study include the large sample size including a large proportion of vegetarians and vegans. Furthermore, the questionnaire was designed to identify dietary groups [2]. Additionally, previous work with this cohort suggests that participants had a high adherence to diet group over time [31]. However, some limitations should be considered when interpreting our findings. Dietary intake was self-reported and could be subject to misreporting, especially regarding unhealthy food items such as SSBs [32]. The generalizability of our results might be limited by the ‘health-conscious’ make up of our cohort and our cohort structure, which is predominantly of white, European descent. It is also possible that some vegetarian and vegan products that are commonly consumed were not captured in our FFQ. However, care was taken to include additional plant-based protein foods in the 2010 FFQ.

## 5. Conclusions

In this large study of British men and women, we compared intakes of major protein-source foods and other food groups in regular meat-eaters, low meat-eaters, poultry-eaters, fish-eaters, vegetarians and vegans. Our results show that meat-eaters and low and non-meat-eaters do not only differ in their meat consumption but in their overall dietary intake; low and non-meat-eaters consume higher amounts of high protein meat alternatives, a wide variety of other plant-based foods as well as lower amounts of refined grains, fried foods, alcohol and SSBs. The dietary intakes consumed by low and non-meat-eaters might explain the lower risk for some diseases in these diet groups and can be used as a real-life guide for future work assessing the health impacts of replacing meat intake with plant-based foods or dietary recommendations.

## Figures and Tables

**Figure 1 nutrients-11-00824-f001:**
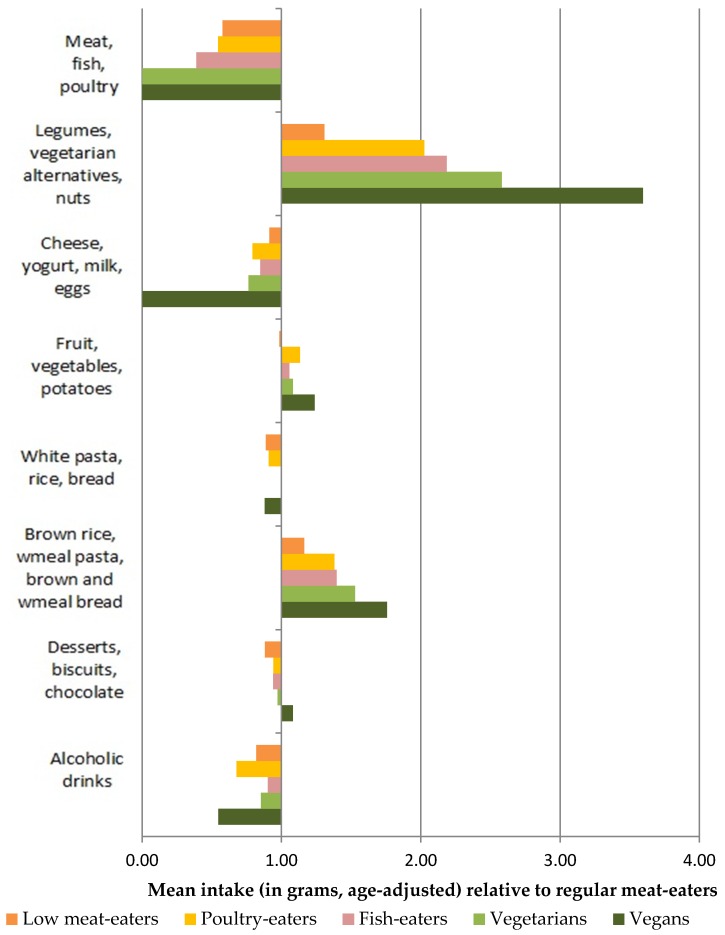
Relative age-adjusted mean (g) consumption of foods in low meat-eaters, poultry-eaters fish-eaters, vegetarians and vegans compared to regular meat-eaters among men.

**Figure 2 nutrients-11-00824-f002:**
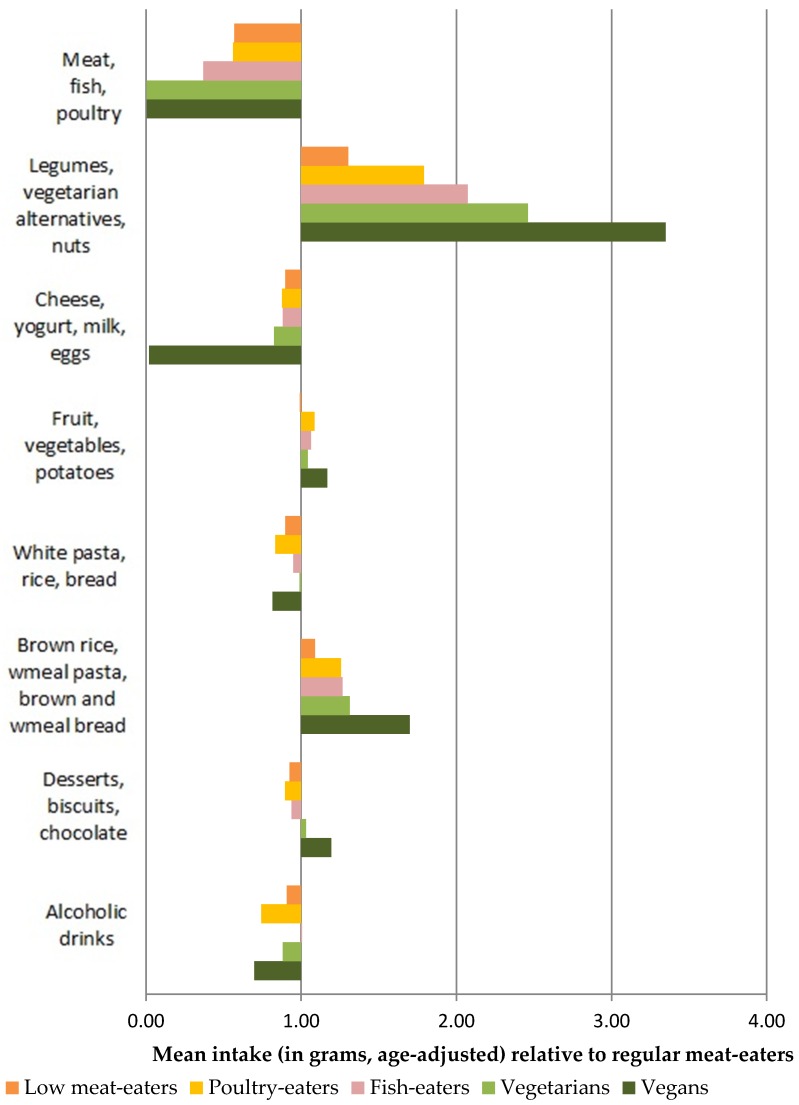
Relative age-adjusted mean (g) consumption of foods in low meat-eaters, poultry-eaters fish-eaters, vegetarians and vegans compared to regular meat-eaters among women.

**Table 1 nutrients-11-00824-t001:** Baseline characteristics of men and women in European Prospective Investigation into Cancer and Nutrition (EPIC)-Oxford by diet group.

Characteristic	Diet Group					
	Regular Meat-Eaters	Low Meat-Eaters	Poultry-Eaters	Fish-Eaters	Vegetarians	Vegans
**Men**	***n* = 2852**	***n* = 880**	***n* = 65**	***n* = 782**	***n* = 1516**	***n* = 269**
**Socio-demographic**						
Age ^a^	63.5 ± 11.6	62.7 ± 11.9	59.2 ± 11.5	58.3 ± 11.2	56.1 ± 11.0	54.2 ± 11.1
Higher education ^b^	1230 (46.2)	420 (51.0)	33 (52.4)	477 (63.1)	817 (56.0)	124 (47.0)
High SES ^b, c^	803 (31.4)	213 (27.4)	10 (16.1)	142 (20.5)	305 (22.7)	54 (22.4)
**Lifestyle and health**						
Current smokers ^b^	385 (13.5)	120 (13.7)	4 (6.2)	95 (12.2)	149 (9.9)	19 (7.1)
Alcohol (grams)	17.6 ± 18.5	14.8 ± 16.4	11.6 ± 16.0	15.5 ± 17.3	14.9 ± 18.6	11.3 ± 16.6
High physical activity level ^b^	380 (14.4)	151 (18.8)	19 (32.8)	142 (19.9)	297 (21.0)	67 (26.2)
Body mass index ^b^	24.9 ± 3.2	23.9 ± 3.0	23.3 ± 3.1	23.5 ± 3.3	23.3 ± 2.9	22.8 ± 3.3
**Diet**						
Total energy kcal	2314 ± 551.8	2088 ± 557.4	2255 ± 616.0	2211 ± 578.5	2192 ± 566.1	2132 ± 633.4
Carbohydrate (%E)	49.7 ± 6.2	52.4 ± 6.2	53.8 ± 7.2	53.0 ± 6.3	54.7 ± 6.4	56.6 ± 8.2
Protein (%E)	16.6 ± 2.4	15.2 ± 2.0	15.1 ± 2.3	14.8 ± 2.2	13.4 ± 1.8	12.5 ± 1.8
Total fat (%E)	31.6 ± 4.6	30.7 ± 4.7	30.9 ± 6.7	30.6 ± 4.9	30.5 ± 5.3	31.0 ± 7.4
**Women**	***n* = 10,145**	***n* = 3770**	***n* = 526**	***n* = 3746**	***n* = 5156**	***n* = 532**
**Socio-demographic**						
Age ^a^	60.4 ± 11.7	59.3 ± 12.0	57.6 ± 12.2	55.7 ± 11.4	52.9 ± 11.2	52.0 ± 11.1
Higher education ^b^	2942 (31.1)	1396 (39.3)	195 (38.8)	1613 (45.0)	2126 (42.8)	208 (40.9)
High SES ^b, c^	2554 (28.8)	805 (24.2)	104 (22.2)	785 (23.7)	1125 (24.4)	81 (16.9)
**Lifestyle and health**						
Current smokers ^b^	985 (9.8)	337 (9.0)	33 (6.3)	312 (8.4)	409 (8.0)	45 (8.5)
Alcohol (grams)	8.3 ± 9.6	7.9 ± 9.7	6.7 ± 8.0	8.6 ± 10.3	7.7 ± 9.9	6.6 ± 10.4
High physical activity level ^b^	896 (10.6)	442 (13.5)	73 (16.0)	501 (15.1)	620 (13.3)	77 (16.0)
Body mass index ^b^	24.3 ± 4.1	23.4 ± 3.7	22.7 ± 3.9	22.6 ± 3.1	22.7 ± 3.5	22.1 ± 2.9
**Diet**						
Total energy kcal	2110 ± 481.2	1900 ± 489.1	1932 ± 501.1	1974 ± 489.5	1940 ± 491.4	1880 ± 519.5
Carbohydrate (%E)	49.3 ± 6.2	52.4 ± 6.5	52.4 ± 7.2	52.9 ± 6.3	55.4 ± 6.5	56.4 ± 7.1
Protein (%E)	17.6 ± 2.5	15.9 ± 2.2	16.6 ± 2.7	15.4 ± 2.2	13.8 ± 1.9	13.0 ± 1.7
Total fat (%E)	32.4 ± 4.9	31.1 ± 5.4	31.0 ± 6.2	30.9 ± 5.5	30.5 ± 5.7	31.0 ± 6.3

Values are presented as mean ± SD or n (%). *x*^2^ test was used to compare the distribution between diet groups for all categorical variables. ANOVA was used to compare the means between the diet groups. The *P*-heterogeneity between diet groups was <0.001 for all variables. SES—socio-economic status, SD—standard deviation, %E—percent energy. Regular meat-eaters were defined as participants who consumed ≥50 grams of total meat (any) per day and low meat-eaters were defined as participants who consumed <50 grams of total meat (any) per day. ^a^ Age at third follow-up ^b^ Numbers may not add to total sample size due to missing responses ^c^ Based on Townsend deprivation index.

**Table 2 nutrients-11-00824-t002:** Major protein-source food intakes in EPIC-Oxford men.

Protein source	Diet Group					
	Regular Meat-Eaters	Low Meat-Eaters	Poultry-Eaters	Fish-Eaters	Vegetarians	Vegans
	*n* = 2852	*n* = 880	*n* = 65	*n* = 782	*n* = 1516	*n* = 269
**Red meat**						
g/day	42.9	18.5	0.1	0.3	0.5	0.7
g/2000 kcal/day ^a^	38.3	19.2	0.1	0.2	0.4	0.5
**Processed meat**						
g/day	15.5	7.4	0.0	0.0	0.1	0.1
g/2000 kcal/day ^a^	13.9	7.5	0.0	0.0	0.0	0.0
**Poultry**						
g/day	36.5	10.5	23.4	-	-	-
g/2000 kcal/day ^a^	32.9	10.9	22.4	-	-	-
**Oily fish**						
g/day	14.3	14.1	19.1	18.1	0.4	0.5
g/2000 kcal/day ^a^	12.6	14.0	17.4	17.4	0.3	0.5
**Non-oily fish**						
g/day	39.1	35.2	38.2	39.4	-	-
g/2000 kcal/day ^a^	35.1	35.4	34.5	37.4	-	-
**Legumes/pulses**						
g/day	30.1	33.2	37.3	42.4	48.4	68.6
g/2000 kcal/day ^a^	26.7	32.5	33.6	39.3	45.7	68.4
**Vegetarian protein alternatives**						
g/day	5.5	13.6	31.3	41.2	50.6	61.0
g/2000 kcal/day ^a^	4.9	12.9	27.2	38.0	47.7	59.6
**Nuts**						
g/day	10.7	13.8	25.1	17.5	20.5	36.6
g/2000 kcal/day ^a^	8.9	12.4	21.9	15.4	17.9	32.6
**Cheese**						
g/day	21.3	22.3	20.6	29.6	33.2	-
g/2000 kcal/day ^a^	18.5	21.0	18.8	26.6	30.1	-
**Yogurt**						
g/day	43.9	42.9	48.9	50.8	40.7	1.2
g/2000 kcal/day ^a^	38.1	41.0	44.4	46.0	36.9	0.8
**Dairy milk**						
g/day	280.6	250.4	203.1	208.9	186.6	0.0
g/2000 kcal/day ^a^	245.3	243.2	183.8	192.1	171.4	-
**Plant milk**						
g/day	6.1	14.1	36.3	31.8	55.4	210.3
g/2000 kcal/day ^a^	5.7	14.6	38.1	30.6	53.8	200.1
**Eggs**						
g/day	19.7	18.3	16.8	20.6	18.6	0.0
g/2000 kcal/day ^a^	17.7	18.1	15.4	19.5	17.7	0.0

All values are age-adjusted for age at follow-up. Regular meat-eaters were defined as participants who consumed ≥50 grams of total meat (any) per day and low meat-eaters were defined as participants who consumed <50 grams of total meat (any) per day. ^a^ Intakes have been standardised to a 2000 kcal daily diet. ANOVA was used to compare the means between the diet groups. The *P*-heterogeneity between diet groups was <0.0001 for all variables.

**Table 3 nutrients-11-00824-t003:** Major protein-source food intakes in EPIC-Oxford women.

Protein Source	Diet Group					
	Regular Meat-Eaters	Low Meat-Eaters	Poultry-Eaters	Fish-Eaters	Vegetarians	Vegans
	*n* = 10,145	*n* = 3770	*n* = 526	*n* = 3746	*n* = 5156	*n* = 532
**Red meat**						
g/day	40.4	17.5	0.0	0.3	0.5	0.5
g/2000 kcal/day ^a^	39.6	19.7	0.0	0.3	0.4	0.4
**Processed meat**						
g/day	12.4	6.4	0.0	0.1	0.0	0.0
g/2000 kcal/day ^a^	12.1	7.1	0.0	0.1	0.0	0.0
**Poultry**						
g/day	40.3	11.5	28.1	0.1	-	-
g/2000 kcal/day ^a^	39.8	13.0	30.9	0.1	-	-
**Oily fish**						
g/day	15.5	14.6	19.0	17.4	0.5	0.6
g/2000 kcal/day ^a^	15.0	15.8	20.6	18.2	0.5	0.6
**Non-oily fish**						
g/day	39.1	34.2	35.9	36.7	-	-
g/2000 kcal/day	38.2	37.4	38.4	38.7	-	-
**Legumes/pulses**						
g/day	28.5	30.9	36.8	39.6	44.7	63.0
g/2000 kcal/day ^a^	27.6	33.1	38.3	41.1	47.5	68.9
**Vegetarian protein alternatives**						
g/day	5.5	13.5	25.8	36.3	47.8	60.0
g/2000 kcal/day ^a^	5.3	14.0	26.9	37.2	50.0	64.9
**Nuts**						
g/day	11.3	14.7	18.6	18.1	19.0	28.8
g/2000 kcal/day ^a^	10.3	14.6	17.9	17.7	18.7	29.7
**Cheese**						
g/day	20.3	19.6	20.3	25.7	28.4	0.3
g/2000 kcal/day ^a^	19.2	20.5	20.6	26.0	29.2	0.3
**Yogurt**						
g/day	58.8	55.6	59.5	60.5	55.0	1.7
g/2000 kcal/day ^a^	56.1	58.5	62.3	61.4	56.9	1.5
**Dairy milk**						
g/day	250.9	221.4	208.6	202.4	187.2	4.9
g/2000 kcal/day ^a^	240.7	237.9	217.8	206.4	194.9	4.4
**Plant milk**						
g/day	11.9	17.6	32.7	36.1	47.7	217.9
g/2000 kcal/day ^a^	12.1	19.1	35.2	38.5	50.5	233.5
**Eggs**						
g/day	18.7	16.6	17.5	18.8	17.0	0.2
g/2000 kcal/day ^a^	18.3	18.0	18.6	19.5	18.0	0.2

All values are age-adjusted for age at follow-up. Regular meat-eaters were defined as participants who consumed ≥50 grams of total meat (any) per day and low meat-eaters were defined as participants who consumed <50 grams of total meat (any) per day. ^a^ Intakes have been standardised to a 2000 kcal daily diet. ANOVA was used to compare the means between the diet groups. The *P*-heterogeneity between diet groups was <0.0001 for all variables.

**Table 4 nutrients-11-00824-t004:** Other food group intakes in EPIC-Oxford men.

Food Group	Diet Group						
	Regular Meat-Eaters	Low Meat-Eaters	Poultry-Eaters	Fish-Eaters	Vegetarians	Vegans	*p* for Difference ^b^
	*n* = 2852	*n* = 880	*n* = 65	*n* = 782	*n* = 1516	*n* = 269	
**Fruit**							
g/day	217	231	299	230	233	277	<0.0001
g/2000 kcal/day ^a^	190	224	262	214	217	281	<0.0001
**Vegetables**							
g/day	255	258	280	297	305	347	<0.0001
g/2000 kcal/day ^a^	227	255	264	277	287	343	<0.0001
**Potatoes—boiled, mashed or jacket**							
g/day	93	76	76	81	81	87	<0.0001
g/2000 kcal/day ^a^	82	74	66	73	75	82	<0.0001
**Potatoes—fried, roasted**							
g/day	31	23	21	23	27	28	<0.0001
g/2000 kcal/day ^a^	27	23	20	21	25	26	<0.0001
**White pasta/noodles**							
g/day	36	33	40	38	37	23	<0.0001
g/2000 kcal/day ^a^	32	32	36	35	35	22	<0.0001
**Wholemeal pasta**							
g/day	14	18	21	24	25	28	<0.0001
g/2000 kcal/day ^a^	12	17	20	22	24	29	<0.0001
**Couscous, bulgur wheat**							
g/day	7	9	13	11	13	16	<0.0001
g/2000 kcal/day ^a^	6	8	11	11	12	16	<0.0001
**White rice**							
g/day	24	20	21	22	21	18	<0.0001
g/2000 kcal/day ^a^	22	20	19	20	20	19	0.0706
**Brown rice**							
g/day	11	14	24	19	19	27	<0.0001
g/2000 kcal/day ^a^	10	14	24	18	18	27	<0.0001
**Pizza**							
g/day	10	10	8	12	13	5	<0.0001
g/2000 kcal/day ^a^	9	10	7	11	13	5	<0.0001
**White bread**							
g/day	27	21	12	22	22	24	<0.0001
g/2000 kcal/day ^a^	23	20	9	19	20	20	0.0002
**Brown bread**							
g/day	20	20	14	19	21	16	0.2225
g/2000 kcal/day ^a^	17	18	12	17	18	15	0.1262
**Wholemeal bread**							
g/day	36	43	54	50	59	71	<0.0001
g/2000 kcal/day ^a^	31	38	45	44	51	63	<0.0001
**Other bread**							
g/day	7	7	6	8	8	8	<0.0001
g/2000 kcal/day ^a^	6	7	6	7	8	8	<0.0001
**Porridge**							
g/day	33	40	43	40	33	44	<0.0001
g/2000 kcal/day ^a^	30	39	45	39	32	44	<0.0001
**Breakfast cereal**							
g/day	33	31	37	32	34	33	0.1723
g/2000 kcal/day ^a^	29	30	34	28	31	30	0.0754
**Cereal bars**							
g/day	12	12	15	12	13	12	0.3626
g/2000 kcal/day ^a^	10	11	13	10	11	10	0.0680
**Chocolate**							
g/day	13	12	18	13	13	11	0.0519
g/2000 kcal/day ^a^	11	11	15	11	11	9	0.0561
**Cake**							
g/day	34	31	32	33	32	25	0.0021
g/2000 kcal/day ^a^	29	29	27	28	28	22	0.0054
**Ice cream**							
g/day	15	12	17	12	13	10	<0.0001
g/2000 kcal/day ^a^	13	12	15	11	12	9	<0.0001
**Milk desserts**							
g/day	53	43	31	45	46	44	0.0012
g/2000 kcal/day ^a^	44	41	26	39	41	35	0.0152
**Soya dessert**							
g/day	1	2	8	7	8	38	<0.0001
g/2000 kcal/day ^a^	1	2	9	7	8	39	<0.0001
**Crisps**							
g/day	1	1	2	1	1	1	0.0002
g/2000 kcal/day ^a^	1	1	1	1	1	1	<0.0001
**Tea**							
g/day	483	461	472	495	468	419	0.0146
g/2000 kcal/day ^a^	435	458	437	470	445	420	0.0983
**Coffee**							
g/day	342	292	183	268	290	204	<0.0001
g/2000 kcal/day ^a^	306	294	164	257	277	195	<0.0001
**Fruit smoothie**							
g/day	133	124	169	153	142	148	0.0029
g/2000 kcal/day ^a^	115	122	144	138	129	141	0.0003
**Fruit squash**							
g/day	70	53	37	48	45	43	<0.0001
g/2000 kcal/day ^a^	59	51	32	40	41	42	<0.0001
**Sugar-sweetened beverages**							
g/day	32	23	39	18	21	23	<0.0001
g/2000 kcal/day	27	23	35	16	19	26	0.0008
**Diet drinks**							
g/day	43	28	58	25	23	34	<0.0001
g/2000 kcal/day	40	28	48	25	22	38	0.0006
**Wine and champagne**							
g/day	104	89	55	89	82	46	<0.0001
g/2000 kcal/day ^a^	90	86	47	80	75	43	<0.0001
**Beer**							
g/day	125	99	100	120	114	79	0.0066
g/2000 kcal/day ^a^	110	98	83	104	105	71	0.0544
**Spirits**							
g/day	4	3	3	2	3	3	<0.0001
g/2000 kcal/day ^a^	4	3	3	2	3	3	<0.0001

All values are age-adjusted for age at follow-up. Regular meat-eaters were defined as participants who consumed ≥50 grams of total meat (any) per day and low meat-eaters were defined as participants who consumed <50 grams of total meat (any) per day. ^a^ Intakes have been standardised to a 2000 kcal daily diet. ^b^ ANOVA was used to compare the means between the diet groups.

**Table 5 nutrients-11-00824-t005:** Other food group intakes in EPIC-Oxford women.

Food Group	Diet Group						
	Regular Meat-Eaters	Low Meat-Eaters	Poultry-Eaters	Fish-Eaters	Vegetarians	Vegans	*p* for Difference ^b^
	*n* = 10,145	*n* = 3770	*n* = 526	*n* = 346	*n* = 5156	*n* = 532	
**Fruit**							
g/day	239	255	263	258	247	268	<0.0001
g/2000 kcal/day ^a^	228	272	282	266	260	294	<0.0001
**Vegetables**							
g/day	296	293	347	333	326	383	<0.0001
g/2000 kcal/day ^a^	288	319	375	349	347	430	<0.0001
**Potatoes—boiled, mashed or jacket**							
g/day	83	69	71	73	76	80	<0.0001
g/2000 kcal/day ^a^	79	74	73	74	78	84	<0.0001
**Potatoes—fried, roasted**							
g/day	25	19	18	19	21	19	<0.0001
g/2000 kcal/day ^a^	24	21	19	19	22	21	<0.0001
**White pasta/noodles**							
g/day	34	31	28	34	34	20	<0.0001
g/2000 kcal/day ^a^	33	33	29	35	36	23	<0.0001
**Wholemeal pasta**							
g/day	14	16	21	21	22	28	<0.0001
g/2000 kcal/day ^a^	13	17	22	22	23	30	<0.0001
**Couscous, bulgur wheat**							
g/day	9	10	12	13	13	16	<0.0001
g/2000 kcal/day ^a^	8	11	13	13	14	17	<0.0001
**White rice**							
g/day	21	18	17	17	17	15	<0.0001
g/2000 kcal/day ^a^	21	19	18	18	18	17	<0.0001
**Brown rice**							
g/day	12	14	19	17	16	23	<0.0001
g/2000 kcal/day ^a^	11	15	21	18	17	26	<0.0001
**Pizza**							
g/day	9	9	8	10	11	6	<0.0001
g/2000 kcal/day ^a^	9	9	9	10	11	6	<0.0001
**White bread**							
g/day	19	14	10	13	17	13	<0.0001
g/2000 kcal/day ^a^	18	15	11	13	17	14	<0.0001
**Brown bread**							
g/day	16	15	13	15	16	15	0.03639
g/2000 kcal/day ^a^	15	15	13	15	16	15	0.0099
**Wholemeal bread**							
g/day	29	32	35	36	38	53	<0.0001
g/2000 kcal/day ^a^	27	32	34	35	38	53	<0.0001
**Other bread**							
g/day	8	8	8	9	9	10	<0.0001
g/2000 kcal/day ^a^	8	8	9	9	9	11	<0.0001
**Porridge**							
g/day	43	45	48	47	40	48	<0.0001
g/2000 kcal/day ^a^	42	49	53	50	43	53	<0.0001
**Breakfast cereal**							
g/day	28	27	27	27	28	24	0.0437
g/2000 kcal/day ^a^	26	28	27	27	28	26	<0.0001
**Cereal bars**							
g/day	10	10	9	10	10	8	<0.0001
g/2000 kcal/day ^a^	10	10	9	10	10	8	0.0002
**Chocolate**							
g/day	13	12	12	12	13	12	0.0015
g/2000 kcal/day ^a^	12	12	12	11	13	12	0.0006
**Cake**							
g/day	27	25	22	24	25	19	<0.0001
g/2000 kcal/day ^a^	25	25	22	24	25	19	<0.0001
**Ice cream**							
g/day	12	10	10	10	11	8	<0.0001
g/2000 kcal/day ^a^	11	11	10	10	11	8	<0.0001
**Milk desserts**							
g/day	55	49	45	48	53	42	<0.0001
g/2000 kcal/day ^a^	50	50	46	46	52	40	0.0004
**Soya dessert**							
g/day	3	5	10	10	12	54	<0.0001
g/2000 kcal/day ^a^	3	5	11	10	13	57	<0.0001
**Crisps**							
g/day	2	2	2	2	2	2	<0.0001
g/2000 kcal/day ^a^	2	2	2	2	2	2	<0.0001
**Tea**							
g/day	507	495	467	512	480	434	<0.0001
g/2000 kcal/day ^a^	499	545	501	542	515	479	<0.0001
**Coffee**							
g/day	289	261	239	251	259	230	<0.0001
g/2000 kcal/day ^a^	286	289	257	265	280	261	0.0004
**Fruit smoothie**							
g/day	108	101	108	113	120	135	<0.0001
g/2000 kcal/day ^a^	101	107	112	113	123	145	<0.0001
**Fruit squash**							
g/day	67	52	51	48	57	50	<0.0001
g/2000 kcal/day ^a^	63	52	51	47	58	56	<0.0001
**Sugar-sweetened beverages**							
g/day	27	20	14	15	20	26	<0.0001
g/2000 kcal/day	26	21	15	15	21	29	<0.0001
**Diet drinks**							
g/day	55	36	29	31	43	16	<0.0001
g/2000 kcal/day	55	41	34	34	48	19	<0.0001
**Wine and champagne**							
g/day	83	74	64	81	69	43	<0.0001
g/2000 kcal/day ^a^	80	80	68	84	73	47	<0.0001
**Beer**							
g/day	25	24	17	28	26	32	0.0020
g/2000 kcal/day ^a^	24	26	18	28	28	35	<0.0001
**Spirits**							
g/day	2	2	1	2	2	2	<0.0001
g/2000 kcal/day ^a^	2	2	2	2	2	2	0.0033

All values are age-adjusted for age at follow-up. Regular meat-eaters were defined as participants who consumed ≥50 grams of total meat (any) per day and low meat-eaters were defined as participants who consumed <50 grams of total meat (any) per day. ^a^ Intakes have been standardised to a 2000 kcal daily diet. ^b^ ANOVA was used to compare the means between the diet groups.

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
