# Peer review of "Comparison of Major Protein-Source Foods and Other Food Groups in Meat-Eaters and Non-Meat-Eaters in the EPIC-Oxford Cohort"

_nutrients, 2019, doi:10.3390/nu11040824_

Round 1
Reviewer 1 Report
General comments
Overall the study is properly described and well-written. The significance and originality of the study was unclear. The author's indicated that this is one of the only studies on this topic in the UK. However, the other study cited by the authors (Bradbury, 2017), was conducted in the UK, had a larger sample size (>199,000), and a more accurate dietary assessment (24h recalls). The authors should provide more information in the introduction on what makes the present study unique and how it will add to the current literature.
Specific comments:
What was the rationale for using 50 g as the cutoff point for regular versus low meat-eaters?
It would be helpful to add total energy intake for each diet group in table 1
Author Response
Reviewer #1 (Comments to the Author):
General comments
1. Overall the study is properly described and well-written. The significance and originality of the study was unclear. The author's indicated that this is one of the only studies on this topic in the UK. However, the other study cited by the authors (Bradbury, 2017), was conducted in the UK, had a larger sample size (>199,000), and a more accurate dietary assessment (24h recalls). The authors should provide more information in the introduction on what makes the present study unique and how it will add to the current literature.
Ø Thank you for your suggestion regarding the introduction. We agree and have highlighted that previous studies were based on small numbers of vegetarians or vegans. The present study has a larger number of vegetarians and vegans than the previous UK study by Bradbury et al 2017. Additionally, we have used an FFQ that incorporates vegetarian alternative foods, adapted for non-meat eaters, whereas the 24h recall used by Bradbury et al was a 24h and FFQ hybrid with fewer questions about vegetarian alternatives. We have now also added a more detailed description in our revised text.
The additional text now reads as follows:
Lines 46-48
Therefore, there is a need for more information on this in studies with a large number of vegetarians to further characterize the food intakes of non-meat-eaters.
Specific comments:
2. What was the rationale for using 50 g as the cutoff point for regular versus low meat-eaters?
Ø We used a 50 gram cut-off for regular versus low-meat eaters for the following reasons: 1) the average consumption of meat is quite low in the EPIC-Oxford cohort (~35 grams in low-meat-eaters, and ~95 grams in regular meat-eaters); 2) the average consumption of red and processed meat in UK adults (as measured by the national diet and nutrition survey) is around ~60 grams; and 3) to be consistent with previous work (see Papier et al 2019, Nutrition & Diabetes 9(1))
3. It would be helpful to add total energy intake for each diet group in table 1
Ø Thank you for this good suggestion. We have now added this information in our revised Table 1.

Reviewer 2 Report
It is evident in the article that human health depends above all on a correct and balanced diet, not so much from being carnivorous, vegetarian or vegan.
Moreover, the number of participants of the Group "regular-meat " is much higher compared to low meat-eaters, poultry-eaters, fish-eaters, vegetarian and vegan.
Furthermore, the authors have not reported in the article the classic biochemical serum parameters that allow to evaluate the health of the participants in the study.
Author Response
Reviewer #2 (Comments to the Author):
1. It is evident in the article that human health depends above all on a correct and balanced diet, not so much from being carnivorous, vegetarian or vegan.
Ø Thank you for your comment. This is an interesting point and we look forward to learning more about the associations between diet groups and health outcomes as more research is conducted.
2. Moreover, the number of participants of the Group "regular-meat " is much higher compared to low meat-eaters, poultry-eaters, fish-eaters, vegetarian and vegan.
Ø We agree and this is the reason we used the regular meat-eaters as the reference group in our comparative analysis. However, the proportion of non-meat eaters in the EPIC-Oxford cohort is high (around 45%), making it one of the largest cohorts of non-meat eaters globally.
3. Furthermore, the authors have not reported in the article the classic biochemical serum parameters that allow to evaluate the health of the participants in the study.
Ø Thank you for your comment. We agree that this would be interesting to examine, however assessing the health of participants was not in the scope of our study.
